# Toxicokinetics of Hydrolyzed Fumonisin B_1_ after Single Oral or Intravenous Bolus to Broiler Chickens Fed a Control or a Fumonisins-Contaminated Diet

**DOI:** 10.3390/toxins12060413

**Published:** 2020-06-21

**Authors:** Gunther Antonissen, Siegrid De Baere, Barbara Novak, Dian Schatzmayr, Danica den Hollander, Mathias Devreese, Siska Croubels

**Affiliations:** 1Department of Pharmacology, Toxicology and Biochemistry, Faculty of Veterinary Medicine, Ghent University, Salisburylaan 133, 9820 Merelbeke, Belgium; Siegrid.DeBaere@ugent.be (S.D.B.); Danica.denHollander@ugent.be (D.d.H.); Mathias.Devreese@ugent.be (M.D.); Siska.Croubels@ugent.be (S.C.); 2Department of Pathology, Bacteriology and Avian Diseases, Faculty of Veterinary Medicine, Ghent University, Salisburylaan 133, 9820 Merelbeke, Belgium; 3Biomin Research Center, Technopark 1, 3430 Tulln, Austria; Barbara.Novak@biomin.net (B.N.); Dian.Schatzmayr@biomin.net (D.S.)

**Keywords:** mycotoxins, toxicokinetics, fumonisins metabolites, feeding trial, broiler chicken

## Abstract

The toxicokinetics (TK) of hydrolyzed fumonisin B_1_ (HFB_1_) were evaluated in 16 broiler chickens after being fed either a control or a fumonisins-contaminated diet (10.8 mg fumonisin B_1_, 3.3 mg B_2_ and 1.5 mg B_3_/kg feed) for two weeks, followed by a single oral (PO) or intravenous (IV) dose of 1.25 mg/kg bodyweight (BW) of HFB_1_. Fumonisin B_1_ (FB_1_), its partially hydrolyzed metabolites pHFB_1a_ and pHFB_1b_, and fully hydrolyzed metabolite HFB_1_, were determined in chicken plasma using a validated ultra-performance liquid chromatography–tandem mass spectrometry method. None of the broiler chicken showed clinical symptoms of fumonisins (FBs) or HFB_1_ toxicity during the trial, nor was an aberration in body weight observed between the animals fed the FBs-contaminated diet and those fed the control diet. HFB_1_ was shown to follow a two-compartmental pharmacokinetic model with first order elimination in broiler chickens after IV administration. Toxicokinetic parameters of HFB_1_ demonstrated a total body clearance of 16.39 L/kg·h and an intercompartmental flow of 8.34 L/kg·h. Low levels of FB_1_ and traces of pHFB_1b_ were found in plasma of chickens fed the FBs-contaminated diet. Due to plasma concentrations being under the limit of quantification (LOQ) after oral administration of HFB_1_, no toxicokinetic modelling could be performed in broiler chickens after oral administration of HFB_1_. Moreover, no phase II metabolites, nor N-acyl-metabolites of HFB_1_ could be detected in this study.

## 1. Introduction

Fumonisins (FBs) are secondary fungal metabolites of *Fusarium verticillioides*, *F. proliferatum* or other *Fusarium* species, that resemble each other structurally [1,2]. Fumonisin B_1_ (FB_1_), B_2_ (FB_2_) and B_3_ (FB_3_) represent the most important of the more than 15 FBs analogues that have been described [3]. FBs can be divided into four main categories, namely A, B, C and P [4]. Since FB_1_ is the most prevalent and toxic analogue, toxicological assessment of FBs has mostly been done for FB_1._ FB_1_ is a diester of propane-1,2,3-tricarboxylic acid (TCA), which has a long aminopolyol side-chain, namely 2-amino-12,16-dimethyl-3,5,10,14,15-pentahydroxyeicosane [3]. FB_2_ and FB_3_ differ structurally from FB_1_ in the number and position of hydroxyl groups, since a free hydroxyl group is lacking on either the C-10 or the C-5 position, as shown in Figure 1 [2]. The primary amino group is known for the biological activity of FBs, as acetylation of FB_1_ to fumonisin A_1_ hampers the cytotoxicity as well as the inhibition of ceramide synthase [3].

FBs reveal distinct structural similarities to sphingolipids, due to their long-chain base backbones. Hence, FBs have the ability to completely inhibit sphinganine N-acyl transferase (ceramide synthase), resulting in a disruption of the ceramide and sphingolipid metabolism [6,7,8]. Blockage of the ceramide synthase enzyme not only leads to the inhibition of sphingolipids synthesis, but also induces an accumulation of free sphinganine (Sa) and sphingosine (So) in tissues, serum and urine [2,4]. This associated increase of the Sa:So ratio, observed in tissues and body fluids after exposure to FBs, is a suitable biomarker of effect in animals and humans [2,9,10]. In most animal species, the liver, as well as the kidneys and the intestinal tract, are the main target organs of FBs toxicity [2,11,12]. Generally, poultry are rather resistant to FBs in comparison to mammals, especially pigs and horses [13,14,15,16,17,18]. Only during the first three days of life (≥125 mg/kg feed) was an increase in mortality after FB_1_ exposure observed in broiler chickens [13], as well as in growing ducks aged 12–14 weeks (20 mg/kg feed) [16]. Feeding a FB_1_-contaminated diet (100–400 mg/kg feed) to broilers resulted in black, sticky diarrhea in the first 2 weeks of life [15]. Besides, broiler chickens fed an FBs-contaminated diet in doses from 100 to 400 mg FB_1_/kg feed for 2 to 3 weeks showed a dose-dependent decrease in feed intake and bodyweight gain [13,15]. No negative effects on performance were observed under experimental conditions, at concentration levels which were below or approached the European maximum guidance level in feed (20 mg FB_1_ + FB_2_/kg feed) [19]. However, it was recently demonstrated that even low to moderate dietary levels of FBs negatively affect enterocyte viability and proliferation, and the production of pro-inflammatory cytokines, and alter the intestinal barrier function, and therefore increase the susceptibility of avian species to important enteric infectious diseases, such as coccidiosis and necrotic enteritis [20,21,22,23]. Furthermore, the mRNA expression of genes encoding for the intestinal cytochrome P450 (CYP450), the drug-metabolizing enzyme CYP1A4, and the drug transporter mechanisms, such as multiple drug resistance protein 1 (MDR1 or P-glycoprotein, P-gp), in broiler chickens were upregulated following exposure to 25 mg FB_1_ + FB_2_/kg feed for 15 days, which can affect the pharmacokinetic (PK) or toxicokinetic (TK) properties of other xenobiotics substrates [24].

Therefore, taking into account that mycotoxin contamination of feed is a continuous feed safety issue, resulting in economic losses during animal production [25], methods for the detoxification of feed have been developed. For FBs, these imply feed processing techniques, such as alkaline cooking (nixtamalization) used during the production of masa and tortillas for human consumption, and the application of mycotoxin binders/modifiers in animal feed. Mycotoxin binders, such as clays, are highly effective against aflatoxins, however, their detoxification capacity wit regards to FBs is rather limited. In contrast, following the application of FB esterase FumD (EC 3.1.1.87), enzymatic degradation results in conversion of FBs to the less toxic partially hydrolyzed FB_1_ (pHFB_1_), or fully hydrolyzed HFB_1_, by cleavage of the tricarballylic acid esters at the C-14 and/or C-15 position. FB esterase FumD is an enzyme of bacterial FB catabolism [26,27], which has recently been commercialized as FUM*zyme*^®^ (BIOMIN GmbH, Getzersdorf, Austria) and authorized by the EU for usage in poultry and pigs. Enzymatic hydrolysis has been shown to be an effective detoxification method in broilers, since it prevents the FB-associated disruption of the sphingolipid metabolism, characterized by a reduced serum and hepatic Sa:So ratio and a counteracting FB-induced intestinal up-regulation of cytokine gene expression [interleukin-8 (IL-8) and IL-10] [28]. Besides, in accordance with different mammalian species [29,30,31], it has been demonstrated that the intestinal microbiota of broiler chickens and turkeys also has a limited capacity to hydrolyze FBs [28,32].

The only known metabolic pathway of FB_1_ is the hydrolyzation (hydrolysis) of its side-chains in vertebrates [11]. HFB_1_ is not only a 10-fold less potent inhibitor of ceramide synthase than FB_1_, but it is also a substrate for ceramide synthase in rat liver microsomes [5,26,31,33,34]. Acetylation of HFB_1_ at the primary amino group, with fatty acids of variable chain lengths, results in the formation of N-acyl-HFB_1_ (NAHFB_1_) metabolites, which are also known to be ceramide analogues. The in vivo formation of NAHFB_1_ has been demonstrated in rats [5]. Furthermore, phase II conjugation processes of FB_1_, e.g., sulphatation as proposed by Hopmans et al. (1997) [35], or glucuronidation, have not yet been identified. However, no data are available on the in vivo TK of HFB_1_, nor on the formation of NAHFB_1_ in chickens [36]. Furthermore, FB_1_ is not absorbed in vitro by human colon adenocarcinoma Caco-2 cells, also suggesting a low oral bioavailability. The oral bioavailability of FB_1_ in chickens is only 0.7% [36]. In contrast, for HFB_1_ a high transcellular passage has been observed in vitro, particularly from the basolateral to the apical side, suggesting that HFB_1_ is effluxed by P-gp [33,37]. Remarkably, this P-gp transport mechanism was upregulated following exposure to the toxic FBs parent molecules, which could affect the TK properties of the HFB_1_ metabolite, as it is a P-gp substrate. Since exposure to HFB_1_ is linked with the hydrolysis of FB_1_ by intestinal enzymes or by a feed additive enzyme, the aim of this study was to investigate not only the TK parameters of HFB_1_ alone, but also the impact of prior exposure to FBs on these parameters.

## 2. Results and Discussion

During the animal experiment, no signs of illness due to FBs or HFB_1_ toxicity were observed in any of the broiler chickens. All birds were alert, showed a normal feed intake and normal droppings, and no regurgitation was observed. The average bodyweight (BW) was similar (*p* = 0.820) in both groups (control and FBs diet), 974 ± 109 and 1036 ± 110 g at day 21, respectively. This matched expectations, since the contaminated feed contained 10.8 mg FB_1_, 3.3 mg FB_2_ and 1.5 mg FB_3_/kg feed, and therefore approached, but was below, the EU guidance level of 20 mg FB_1_ + FB_2_/kg [19]. Additionally, HFB_1_ is generally known to be less toxic than its parent toxin FB_1_ [26,27], and poultry species are known to be able to tolerate high doses of FB_1_, especially regarding performance response, with no effect on growth of broiler chickens up to 75–100 mg FB_1_/kg feed [15].

To the authors’ knowledge, for the first time, the TK properties of HFB_1_ were determined in broiler chickens, both after feeding a control and FBs-contaminated diet. In pigs, biotransformation of FB_1_ into pHFB_1_, and further to HFB_1_, by their digestive microbiota and liver has been observed after being fed an FBs-contaminated diet containing 45 mg FB_1_, 8.6 mg FB_2_ and 4.6 mg FB_3_/kg for 10 days, together with a persistence of low levels of pHFB_1_ in most organs for several days [30]. However, only low concentrations of pHFB_1a_ and pHFB_1b_ have been detected in the plasma of broiler chickens, fed an FBs-contaminated diet with 10 mg FBs/kg for 14 days [28]. The formation of pHFB_1a_, pHFb_1b_ and HFB_1_ after FB_1_ exposure has been described in turkeys [32], monkeys [29] and rats [31] as well. Accordingly, in this study, only traces (>limit of quantification; LOQ) of pHFB_1b_ (0.94–1.34 ng/mL and 0.93–2.43 ng/mL, respectively) were detected in a few samples of broiler chickens fed the FBs-contaminated diet, and subsequently administered an IV or PO bolus of HFB_1_, respectively. Although, the presence of pHFB_1a_ was detected in some samples of broilers fed the FBs-contaminated diet (>limit of detection; LOD), it could not be quantified (<LOQ). No pHFB_1a_ or pHFB_1b_ was detected in chickens fed the control diet. Low levels of FB_1_ were observed in chickens fed the FBs-contaminated diet, with average background levels at all timepoints of 4.89 ± 5.91 ng/mL and 2.67 ± 3.70 ng/mL, in broiler chickens administered HFB_1_ IV and PO, respectively. No FB_1_ was detected in the plasma of animals fed the control diet. Consequently, it can be concluded that the observed pHFB_1_ in the plasma of animals fed the FBs-contaminated diet resulted from hydrolysis of the FB_1_ present in the diet. However, the intestinal microbiota of broiler chickens and turkeys seems to have a rather limited capacity to hydrolyze FBs [28,32]. HFB_1_ initiated lesser inflammatory responses than FB_1_ in an in vitro co-culture model of porcine intestinal epithelial and immune cells [38]. In vivo toxicity of the intermediate, pHFB_1_, has only been investigated in rats so far, resulting in no observed in vivo toxicity [31]. On the other hand, hepatic and renal lesions, in terms of single-cell necrosis, mitosis, apparent collapse of the centrilobular parenchyma of the liver and single-cell necrosis of the outer medulla, and cytoplasmic vacuolation of the kidney, have been reported in rats fed nixtamalized feed containing HFB_1_ [39], while no hepatotoxicity or pathological changes of the liver, brain, heart, kidneys, thymus or mesenteric lymph nodes examined in mice fed purified HFB_1_ have been detected [40]. This different outcome could probably be explained by the presence of residual, partially hydrolyzed or masked FB_1_, following the process of nixtamalization [5]. However, no significant influence of HFB_1_ on the intestinal lesion score of the villi morphology in piglets exposed to HFB_1_ has been observed [41].

No N-acyl or phase II metabolites of HFB_1_ were detected in any of the samples in this study. N-acyl-metabolites of HFB_1_ were highly cytotoxic in human colonic HT29 cells, and were also characterized as a potent inhibitor of ceramide synthase [34]. HFB_1_ acylation has been evaluated in rats, and although N-acyl-metabolites were detected, no toxicity was observed in vivo [5]. While N-acylation of FB_1_ and HFB_1_ occurred in human cell lines and in rats [33,34], the occurrence of these metabolites in avian species has not yet been observed, matching the absence of these metabolites in this study.

Comparative HFB_1_ plasma concentration–time profiles, following IV HFB_1_ administration to broiler chickens fed either a control diet or a FBs-contaminated diet, are shown in Figure 2. Visual inspection of the goodness-of-fit plots of the individual, model-predicted concentrations (IPRED) versus the observed HFB_1_ plasma concentrations (C_obs_) revealed an appropriate structural model for most individuals (Appendix A). The QQ-plots of the conditionally weighted residuals of C_obs_ demonstrated the normal distribution of the weighted residuals (Appendix A). The addition of the covariate experimental diet (control versus FBs-contaminated) did not significantly improve the -2 log likelihood-ratio (2LL) of any of the fixed effect parameters’ volume of distribution of the central compartment (V_c_), volume of distribution of the peripheral compartment (V_p_), total body clearance (Cl), or intercompartmental flow (Q), and was therefore not retained in the final model.

The toxicokinetic parameters of HFB_1_ following IV administration are shown in Table 1, and are described by a two-compartment model. The mean plasma concentration of HFB_1_ in broiler chickens fed an FBs-contaminated diet for 2 weeks following IV administration (plasma concentration at time 0; C_0_) was 339.59 ng/mL. The total exposure of HFB_1_ in broiler chickens after IV administration, from timepoint 0 to infinity (area under the plasma-concentration time curve from time 0 to infinity; AUC_0-inf_), was 76.26 ng·h/mL. Regarding volume of distribution (Vd), values of 3.68 L/kg and 5.04 L/kg have been determined for the central and peripheral compartments, respectively, thus resulting in a Vd of 8.7 L/kg at steady state (V_ss_). Accordingly, pigs given a single HFB_1_ IV bolus showed a Vd of 11.0 L/kg, in a one-compartmental model [42]. Furthermore, a short distribution half-life (T_1/2α_) of 0.09 h, followed by a longer elimination half-life (T_1/2β_) of 0.69 h, was revealed. In barrows given a single IV bolus, of 0.056 mg/kg BW, of HFB_1_, a T_1/2α_ of 0.05 h and a T_1/2β_ of 1.01 ± 0.80 h was observed [42]. The mean residence time (MRT) of HFB_1_ after IV administration to broiler chickens was 0.53 h, and the elimination rate constant (k_e_) was 4.4 h^−1^. In comparison to pigs, with a total body clearance of 6.7 L·kg/h in barrows after IV HFB_1_ administration, broilers seem to have a higher total body clearance (16.39 L/kg·h). The latter could concur with the oral bioavailability of parental FB_1_ being higher in swine than in poultry, with values of 3.1% [42] and 0.7% [36], respectively, and with poultry being rather resistant to FBs toxicity when compared to pigs or other mammals [13,14,15,16,17,18].

Plasma concentrations of HFB_1_ after oral administration were only above the LOQ in a limited number of samples, and are shown in Figure 3. Consequently, no TK modelling could be performed on these data. Still, the rather fast appearance of HFB_1_ in the systemic circulation after oral administration indicates that the ingested mycotoxin is mainly absorbed in the proximal part of the small intestines.

The T_max_ of HFB_1_ (10 min) after oral HFB_1_ administration in the FBs-fed group was reached remarkably fast, when compared to studies of oral FB_1_ intake in laying hens, turkeys and ducks, where respective T_max_ values of 60 min, 180 min and 60–120 min were observed [43].

Previously, a disturbance of intestinal homeostasis of chickens fed an FBs-contaminated diet (18.6 mg FB_1_ + FB_2_/kg feed) was demonstrated [21]. In another study, broiler chickens were either fed an FBs-contaminated (8.70 mg FB_1_/kg, 4.15 mg FB_2_/ kg and 1.44 mg FB_3_/kg feed) or a control diet, resulting in FB_1_ influencing the expression of intestinal P-gp [24]. With HFB_1_ being a substrate of this receptor [24], another aim of this research was to examine whether prolonged FB_1_ exposure would influence the uptake of its metabolite HFB_1_ in the intestines of broiler chickens. Regarding the difference in plasma concentrations between the FBs and the control diet fed groups, after oral administration of HFB_1_, in this study, one could presume that a prolonged FBs exposure actually does influence the intestinal uptake of HFB_1_. But as mentioned before, the quantity of plasma concentration measurements above the LOQ, after oral HFB_1_ administration, was insufficient to allow the drawing of conclusions from those results. The latter is also held responsible for the fact that no calculation of oral bioavailability of HFB_1_ could be made. Nevertheless, a high transcellular passage, particularly from the basolateral to the apical side, was revealed for HFB_1_ in human colon adenocarcinoma Caco-2 cells, suggesting that HFB_1_ is effluxed by P-gp [33,37]. Since an increased expression of multi-drug resistance protein 1 (MDR1), encoding for P-gp, could be observed in the intestine of broiler chickens fed an FBs-contaminated diet [24], oral bioavailability of HFB_1_ could vary, leaving one with the need for more research on the oral bioavailability of HFB_1_ in avian species. In broiler chickens, the oral bioavailability of FB_1_, being mainly negatively charged in the duodenum and jejunum due to the pH which therefore limits oral absorption by passive non-ionic transcellular diffusion [43], is known to be rather low [44]. Similarly, in this study only very low plasma levels of HFB_1_ were detected in broiler chickens following oral administration, suggesting a very low oral bioavailability. That said, in rats, the bioavailability of HFB_1_ was thought to be greater than that of FB_1_, since about 2.5-fold greater amounts of ^14^C-HFB_1_, compared to ^14^C-FB_1_, were excreted in urine after oral administration of 0.69 µmol ^14^C-FB_1_ and ^14^C-hydrolyzed FB_1_/kg BW [45], increasing the likelihood that HFB_1_ can be absorbed by the gut, and in the meantime allowing the subsequent metabolism of N-acyl-HFB_1_ and the enhancing of its possible toxicological effects.

## 3. Conclusions

While HFB_1_ is shown to be less toxic than its parental toxin FB_1_, its toxicokinetic pathways in avian species are still widely unexplored. In this study, toxicokinetic parameters following administration of a single IV bolus of HFB_1_ to broiler chickens have been determined (V_c_, V_p_, V_ss_, Cl, Q, AUC_0-inf_, C_0_, k_e_, MRT, T_1/2α_ and T_1/2β_). The plasma concentrations of HFB_1_ after oral administration were only above the LOQ in a limited number of samples. Consequently, no TK modelling could be performed on these data. Traces of pHFB_1b_ (0.94–1.34 ng/mL and 0.93–2.43 ng/mL, respectively) have been detected in a few samples of broiler chickens fed the FBs-contaminated diet, and subsequently administered an IV bolus or PO bolus of HFB_1_, respectively. Using UPLC-HRMS, no phase II metabolites or N-acyl metabolites of HFB_1_ could be detected in any of the samples.

## 4. Materials and Methods

### 4.1. HFB_1_ for Animal Trial

HFB_1_ for oral (PO) and intravenous (IV) administration was prepared as previously described [31] with some minor changes. HFB_1_ for PO administration was obtained by enzymatic treatment of *F. verticillioides* culture material with the FB carboxylesterase FumD (80 U enzyme/g culture material) for 30 min, and subsequently purified and lyophilized [31]. Per 250 g of culture material, 2550 µmol HFB_1_ was obtained.

HFB_1_ for IV administration was prepared by alkaline hydrolysis of FB_1_ analytical standard (Biopure, Tulln, Austria). Therefore, 5 mL of FB_1_ solution (7.4 mg/mL water) was hydrolyzed with 4 mL of 2.5M NaOH by shaking for 7.5 h. After completing the conversion, the formed HFB_1_ was recovered by liquid–liquid extraction (eight times with acetonitrile (ACN)). The achieved HFB_1_ solution was purified over an SPE C18 column to remove salts and tricarballylic acids, and subsequently diluted in a 0.9% NaCl solution up to a concentration of 4.18 mg HFB_1_/mL, with traces of pHFB_1_a (0.0045 mg/mL). HFB_1_ for PO and IV administration was stored at 2–8 °C.

### 4.2. Feed Preparation and Experimental Diets

A commercial broiler starter diet was fed to all chickens (Vanden Avenne, Ooigem, Belgium) during the first week of the experiment. Later, this diet is referred to as control diet. From day 8 until day 24 chickens were fed either the control diet without mycotoxins [commercial broiler grower diet (Vanden Avenne)], or a grower diet experimentally contaminated with FBs. None of the diets contained a mycotoxin detoxifier. Screening of the control starter and grower feeds for mycotoxin contamination was executed by a liquid chromatography tandem mass spectrometer (LC-MS/MS) method, as described by Monbaliu et al. [46]. The levels of all tested mycotoxins were below the decision limit (CCα) in the control starter and grower diets. More specifically, the CCα values of FB_1_, FB_2_ and FB_3_ were 58, 45 and 42 µg/kg feed, respectively.

To produce the grower diet experimentally contaminated with FBs, FBs culture material was mixed with 500 g of control feed. Lyophilized FBs culture material of *F. verticillioides* (M-3125) [47] (7.37 g FB_1_/g and 2.93 g FB_2_/g) was obtained from Biopure, Romer Labs Diagnostic GmbH (Tulln, Austria) and stored at 2–8 °C. This premix was then blended with 5 kg of control feed to ensure a homogeneous distribution of the mycotoxins. The premix was blended for 20 min into the total quantity of feed necessary for the trial. To test the homogeneity of FBs in feed, samples were taken at three different points in the batch and analyzed for FBs as described for the control diets. The FBs-contaminated diet contained 10.8 mg FB_1_, 3.3 mg FB_2_ and 1.5 mg FB_3_/kg feed.

### 4.3. Animal Experiment

A total of 16 one-day-old broiler chickens (Ross 308) were obtained from a commercial hatchery (Vervaeke-Belavi, Tielt, Belgium), and were fed a starter diet without mycotoxins during the first week. Subsequently, animals were randomly divided into two experimental groups of eight animals (4 male/4 female). While one group was being fed a control grower diet from day 8 onwards, the other group was given a FBs-contaminated grower diet. An 18 h/6 h light/darkness program was applied. Feed and drinking water were provided ad libitum.

A toxicokinetic (TK) study of HFB_1_ was performed following a two-way cross-over design. Each broiler chicken was administered a bolus of HFB_1_ (1.25 mg/kg bodyweight) either orally (PO) or intravenously (IV) (vena cutanea ulnaris superficialis or wing vein). At day 21, four chickens of each group were administered HFB_1_ IV, and the four other animals received HFB_1_ PO. After a wash out and recovery period of two days, the protocol was repeated at day 24 in a cross-over design. Animals were feed-deprived overnight (8 h) prior to, and until 3 h after, HFB_1_ administration. After each bolus dosing, blood was collected into heparinized tubes by direct venipuncture of the leg vein (vena metatarsalis plantaris superficialis) before (0 h) and at different time points after HFB_1_ administration: 5 min, 10 min, 20 min, 30 min, 40 min, 50 min, 1 h, 1.5 h, 2 h, 3 h, 4 h, 6 h and 9 h post-administration (p.a.). Blood samples were centrifuged (2851× *g*, 10 min, 4 °C) and plasma was stored at ≤−15 °C until analysis.

The animal experiment was approved by the Ethical Committee of the Faculty of Veterinary Medicine and the Faculty of Bioscience Engineering of Ghent University (EC 2015/10, approval date: 9 March 2015).

### 4.4. Plasma Fumonisins Analysis: FB_1_, FB_2_, FB_3_, HFB_1_, pHFB_1_a, pHFB_1_b, Phase II Metabolites and N-Acyl Metabolites

FB_1_, and its partially hydrolyzed metabolites pHFB_1a + b_ and hydrolyzed metabolite HFB_1_, in plasma, were determined using a validated sensitive and specific UPLC-MS/MS method as described by De Baere et al. (2018) [48].

#### 4.4.1. Preparation of Standard Solutions

Stock solutions of FB_1_, FB_2_ and FB_3_ (1 mg/mL) were made in water/ACN (50/50, *v*/*v*) and kept at 2–8 °C. Working solutions of 10 µg/mL, 1 µg/mL and 0.1 µg/mL were made by adequate dilution of the stock solution in water/ACN (50/50, *v*/*v*). The standard mixture solution contained FB_1_, HFB_1_, pHFB_1_a and pHFB_1_b and was adequately diluted in water/ACN (50/50, *v*/*v*) in order to prepare working solutions necessary for the preparation of calibrator and quality control (QC) samples. A working solution of 1 µg/mL of ^13^C_34_-FB1 was made in water/ACN (50/50, *v*/*v*) to obtain the internal standard (IS). All working solutions were kept at 2–8 °C.

#### 4.4.2. Plasma Sample Pre-Treatment

To 100 µL of plasma 12.5 µL of the IS working solution (1 µg/mL) was added. Then, after vortex mixing the sample was transfered onto an Ostro^TM^ 96-well plate. Next, 300 µL of 1% formic acid (FA) in ACN was added, after which the sample was aspirated 3 times in order to stimulate protein precipitation. By the application of a vacuum (67.7 kPa) for 10 min, the sample was run through the 96-well plate. An aliquot of 2.5-µL was injected onto the LC-MS/MS instrument.

#### 4.4.3. UPLC-MS/MS Method for Quantification

Briefly, the LC system was composed of an Acquity UPLC H-Class Quaternary Solvent Manager and Flow-Through-Needle Sample Manager with temperature-controlled tray and column oven from Waters (Zellik, Belgium). Chromatographic separation was accomplished on an Acquity UPLC HSS T3 column (100 mm × 2.1 mm i.d., dp: 1.8 µm) (Waters, Zellik, Belgium) combined with an Acquity HSS T3 1.8 μm Vanguard pre-column (Waters, Zellik, Belgium). The mobile phase A was composed of 0.3% FA and 10 mM NH_4_FA in water, and the mobile phase B was ACN. A gradient elution of 0–0.5 min (90% A, 10% B), 5.5 min (linear gradient to 90% B), 5.5–7.5 min (10% A, 90% B), 7.7 min (linear gradient to 90% A) and 7.7–10.0 min (90% A, 10% B) was performed. The flow-rate was set at 0.4 mL/min. The temperatures of the column oven and autosampler tray were adjusted to 40 °C and 8 °C, respectively. The UPLC column effluent was coupled to a Xevo TQ-S^®^ MS/MS system, supplied with a positive electrospray ionization (ESI) probe (all from Waters). A divert valve was utilized and the UPLC effluent was routed to the mass spectrometer from 2.5 to 4.5 min. Optimization of instrument parameters was obtained by directly infusing working solutions of 1 µg/mL of FB_1_, FB_2_, FB_3_, the IS and a diluted standard mixture solution, which contained FB_1_, HFB_1_, pHFB_1_a and pHFB_1_b at concentrations of 0.5, 0.86, 1.43 and 2.5 µg/mL, respectively, combined with the mobile phase (50% A, 50% B, flow-rate: 200 µL/min). The flow-rate was set at 10 µL/min. MS/MS acquisition was operated in the multiple reaction monitoring (MRM) mode [48].

#### 4.4.4. UPLC-HR-MS Analysis for Identification

To determine potential HFB_1_ phase II and N-acyl metabolites in plasma samples of broiler chickens, an Acquity I-Class UPLC interfaced to a Synapt G2-S*i* HDMS instrument (Waters, Zellik, Belgium) was used [48].

### 4.5. Toxicokinetic Modelling

Plasma concentration–time data were analyzed with a nonlinear mixed-effects modelling approach using quasi-random parametric expectation maximization (QRPEM) as an estimation method in Phoenix NLME^®^ (Certara, Cary, NC, USA). The structural toxicokinetic model for the IV data was a two-compartmental model with first order elimination as shown in Equation (1).
(dA_1_/dt) × 1/V_c_ = −Cl × C − Q × (C − C2)(dA_2_/dt) × 1/V_p_ = Q × (C − C2)(1)
where dA_1_/dt is the rate of the decrease of the amount of mycotoxin in the plasma or central compartment, dA_2_/dt is the rate of decrease of the amount of mycotoxin in the peripheral compartment, V_c_ is the volume of distribution of the central compartment, V_p_ is the volume of distribution of the peripheral compartment, Cl is total body clearance, Q is the intercompartmental flow and C and C2 are the concentrations of the toxin in the central and peripheral compartments, respectively.

Interindividual variability was expressed using an exponential error model according to Equation (2):P*_i_* = θ_P_ × e^η^_P*i*_(2)
where P_i_ is the parameter in the *i*th bird, θ_P_ is the typical value of the parameter in the population, and η_Pi_ is a random variable in the *i*th bird, with a mean of zero and a variance of ω². Interindividual variability is reported as ω. Residual variability (ε), with a mean of zero and a variance of σ², was best described with a multiplicative error model in Equation (3).
C_obs_ = C_pred_ × (1 + ε)(3)
where C_obs_ is the observed concentration for the individual and C_pred_ is the model-predicted concentration plus the error value (ε).

Structural and error model selection was guided by visual inspection of goodness-of-fit plots (e.g., observed vs predicted plasma concentrations, weighted residuals versus predicted concentrations, and weighted residuals versus time), –2LL, Akaike information criterion (AIC) and Bayesian information criterion (BIC) as well as precision of the parameter estimates. The models were chosen based on the smaller values of –2LL, AIC and BIC, the better precision of estimates, and the superior goodness-of-fit plots.

The evaluated covariate was the prior feeding with FBs-contaminated feed versus feeding with uncontaminated control feed (categorical variable). A stepwise forward–backward process was used to evaluate whether inclusion of the covariates significantly improved the model fit using a –2LL test. A decrease in –2LL with a *p*-value < 0.01 was considered significant for addition, and *p* < 0.001 for exclusion, of the covariate.

The following fixed effect parameters were determined: V_c,_ V_p_, Cl and Q. The computed secondary parameters were: C_0_, AUC_0-inf,_ K_e_, MRT, T_1/2α_ and T_1/2β_.

## Figures and Tables

**Figure 1 toxins-12-00413-f001:**
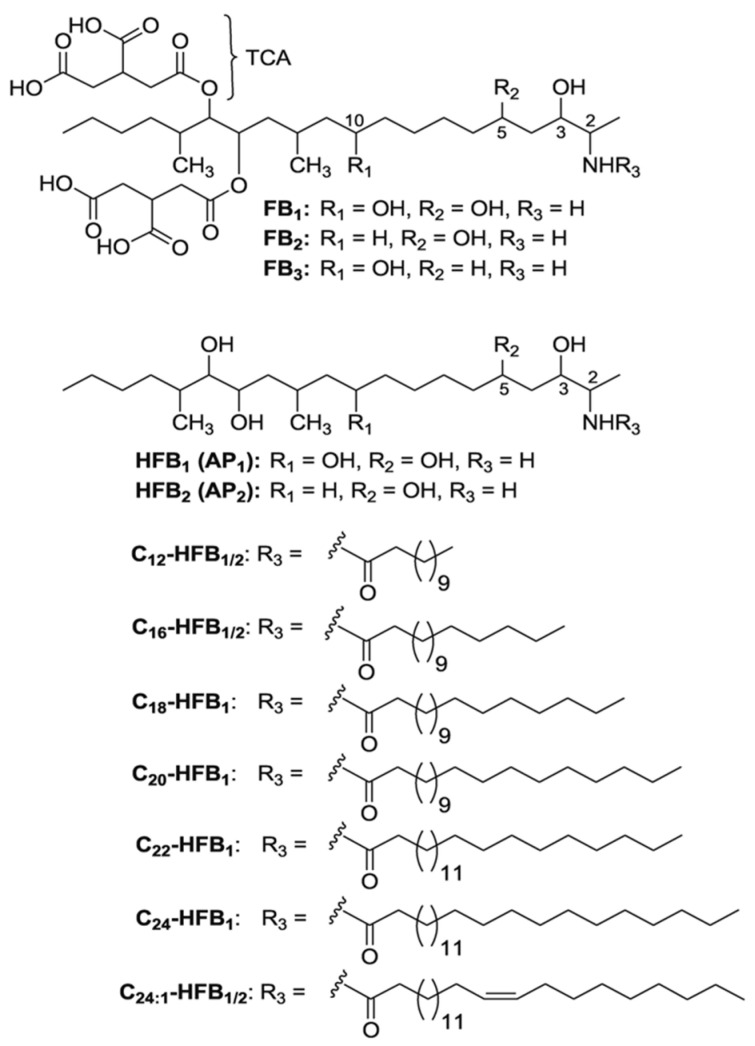
Structures of the fumonisins FB_1_, FB_2_ and FB_3_ with tricarballylic acid side-chains (TCA), and of the hydrolyzed fumonisins HFB_1_ and HFB_2_ (or aminopentols, AP_1_ and AP_2_, respectively) and the corresponding N-acyl-derivatives [5]. Copyright © 2007 WILEY-VCH Verlag GmbH & Co. KGaA, Weinheim.

**Figure 2 toxins-12-00413-f002:**
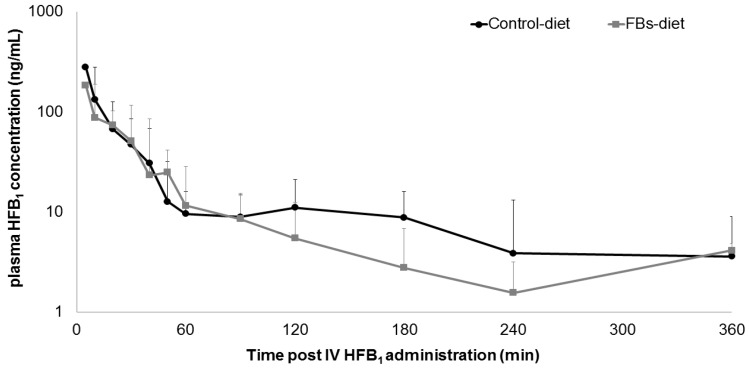
Comparative plasma concentration−time profile of HFB_1_ after intravenous (IV) administration of 1.25 mg HFB_1_/kg bodyweight to broiler chickens fed either a control diet (*n* = 8) or a fumonisins (FBs)-contaminated diet (10.8 mg FB_1_, 3.3 mg FB_2_ and 1.5 mg FB_3_/kg feed, *n* = 8) for two weeks. Values are presented as mean + standard deviation (SD).

**Figure 3 toxins-12-00413-f003:**
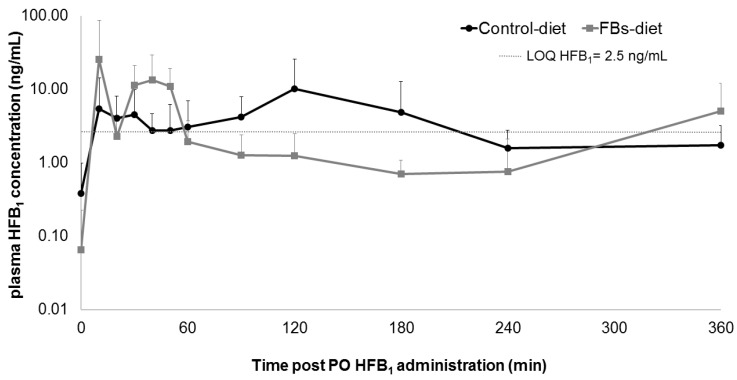
Comparative plasma concentration−time profiles of HFB_1_ after oral (PO) administration of 1.25 mg HFB_1_/kg bodyweight to broiler chickens fed either a control diet (*n* = 8) or a fumonisins (FBs)-contaminated diet (10.8 mg FB_1_, 3.3 mg FB_2_ and 1.5 mg FB_3_/kg feed, *n* = 8) for two weeks. Values are presented as means + standard deviatie (SD).

**Table 1 toxins-12-00413-t001:** Population toxicokinetic results of intravenous (IV) administration of HFB_1_ (1.25 mg/kg BW) to broiler chickens [*n* = 16, 8 animals fed a control diet and 8 animals fed a fumonisins (FBs)-contaminated diet prior HFB_1_ administration].

Θ	Tvθ	CV (%)	ω
V_c_ (L/kg)	3.68	23.47	0.095
V_p_ (L/kg)	5.04	41.47	0.009
V_SS_ (L/kg)	8.72	28.75	/
Cl (L/kg·h)	16.39	12.67	0.126
Q (L/kg·h)	8.34	41.37	0.391
AUC_0-inf_ (ng·h/mL)	76.26	12.67	/
C_0_ (ng/mL)	339.59	23.47	/
K_e_ (1/h)	4.45	26.27	/
MRT (h)	0.53	31.56	/
T_1/2α_ (h)	0.09	33.26	/
T_1/2β_ (h)	0.69	40.41	/

Θ: fixed effect parameter; Tvθ: population typical value of the fixed effect parameter; CV: coefficient of variation; ω: variance of the interindividual variability (only for fixed parameters). Addition of the covariate experimental diet (control versus FBs-contaminated) did not significantly improve the –2 log likelihood (–2LL) of any of the fixed effect parameters, and was therefore not retained in the final model. V_c_: volume of distribution of the central compartment; V_p_: volume of distribution of the peripheral compartment; V_ss_: volume of distribution at steady state; Cl: total body clearance; Q: intercompartmental flow, AUC_0-inf_: area under the plasma concentration–time curve from time 0 to infinity; C_0_: plasma concentration at time 0 following IV administration; K_e_: elimination rate constant; MRT: mean residence time; T_1/2α_: distribution half-life, and T_1/2β_: elimination half-life.

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
