# Peer review of "Toxicokinetics of Hydrolyzed Fumonisin B_1_ after Single Oral or Intravenous Bolus to Broiler Chickens Fed a Control or a Fumonisins-Contaminated Diet"

_toxins, 2020, doi:10.3390/toxins12060413_

Round 1

Reviewer 1 Report

Please see the attached PDF file

Author Response

Point 1: The use of HFB1 and the use of “phase II products” is for me not clear. Does HFB1 refer to the form that has no tricarballyic acid, AP1 (aminopentol)? What are phase II products? Glucuronic adducts and sulfate products? If yes, please define them.

Response 1: Indeed, HFB1 refers to the form that has no tricarballylic acid side-chains. Aminopentol and hydrolyzed FB1 are synonyms. This has been added in the caption of Fig. 1 in the revised version of the manuscript. Phase II products are among others resulting from acetylation, methylation, glycine conjugations and glutathione conjugations, and include indeed glucuronic acid adducts and sulfate conjugates. However, as stated in the article and references [10] and [35], the only known metabolic pathway of FB1 is hydrolyzation of its side-chains and phase II conjugations with sulfate or glucuronic acid are not yet identified for FB1, when comparing to other mycotoxins (e.g. DON or ZEN).

Point 2: Authors declare, that only FB esterase is the enzyme that is able to catalyze the FB1  HFB1 degradation, and this enzyme is present in the fowl feed occasionally (FUMzyme), as an additive enzyme. Accepting this, the in vivo hydrolysis of FB1 in chickens is only minor, as characterizes by ref [27]. However, if the HFB1 is detected in chicken plasma indeed, it can be of even TWO sources (1. hydrolysis by intestinal microbiota and 2. also as an effect of dietary additive enzyme and its hydrolytic activity). Question 1: Was this the exact reason why a FB1-free and a FB1-added feed was also tested? If yes, please explain this clearly.

Response 2: We thank the reviewer for this critical remark. In an earlier study (Antonissen et al., 2017), we observed how FBs up-regulate the expression of P-glycoprotein (P-gp) in the jejunum of broiler chickens fed a FBs contaminated diet, which could affect the toxicokinetic parameters of HFB1, since HFB1 has been reported as a P-gp substrate. Since, exposure to HFB1 is linked with hydrolysis of FB1 by intestinal enzymes or by a feed additive enzyme, the aim of this study was to investigate not only the TK parameters of HFB1 alone, but also the impact of prior exposure to FBs on these parameters.

Point 3: How can it be explained that in Figure 2, higher concentrations values are found at 3 timepoints post-IV HFB1 administration in the control group? Even if FBs diet’s FB1 content is minimally converted to HFB1 (as stated by ref 27 and the Authors), at least similar values should be found for the control and the FBs diet. But here de Y axis in a Log scale, indicating drastic differences at all three timepoints. Please explain this. The same explanation is needed for Figure 3.

Response 3: We agree with the reviewer that the mean plasma concentrations at these 3 time points are higher. However, it should be mentioned that the standard deviations are quite large, as shown in Fig. 2. Actually, the differences between the mean plasma concentrations of the FBs-fed broilers compared to the control diet fed broilers (either post IV or post PO administration) are not statistically significant, taking these SD into account. Therefore, one cannot draw conclusions on any differences between the two groups.

Point 4: L193-194:Still, the rapid appearance of HFB1 in the systemic circulation after oral administration indicates that the ingested toxin is mainly absorbed in the proximal part of the intestinal tract” - can you please define de intestinal tract more exactly.

Response 4: The proximal part of the intestines would be the proximal part of the small intestines, namely the duodenum and jejunum, and presumably less the ileum. This information was added to the revised version of the manuscript.

Point 5: L250: Authors in their study used a commercial broiler diet, as stated at L250. Was it tested or proven, whether this diet was free of any mycotoxin binders? This is crucial from the viewpoint of the FBs diet, which was prepared by the combination of this commercial mix.

Response 5: The commercial broiler diet was free of mycotoxin binders. This has now been added in the revised version of the manuscript. Also, the control diet was analyzed for the presence of mycotoxins. This was mentioned in the manuscript in section 4.2.:” Screening of the control starter and grower feeds for contamination with mycotoxins was performed by a LC-MS/MS method, as described by Monbaliu et al [46]. The levels of all tested mycotoxins were below the decision limit (CCα) in the control starter and grower diets. More specifically, the CCα of FB1, FB2 and FB3 was 58, 45, 42 µg/kg feed, respectively. “

Point 6: Minor comments

Response 6: Please find the revisions in the revised word document of the research article.

Reviewer 2 Report

The manuscript entitled “Toxicokinetics of hydrolyzed fumonisin B1 in broiler chickens fed a control or a fumonisins contaminated diet” is built upon previous work conducted by the same research group. In this work, authors investigate the TK 94 parameters of HFB1 both after feeding a FBs contaminated diet and a control diet. In my opinion, the work does not reach the minimum standards of quality and novelty to deserve publication in Toxins. The study is in general ill designed – and information about the experimental setup contradictory.

Author Response

Point 1: The manuscript entitled “Toxicokinetics of hydrolyzed fumonisin B1 in broiler chickens fed a control or a fumonisins contaminated diet” is built upon previous work conducted by the same research group. In this work, authors investigate the TK parameters of HFB1 both after feeding a FBs contaminated diet and a control diet. In my opinion, the work does not reach the minimum standards of quality and novelty to deserve publication in Toxins. The study is in general ill designed – and information about the experimental setup contradictory.

Respond 1: We would like to thank the reviewer for taking the time to review our manuscript. However, the previous work described the analytical method, while this paper describes the toxicokinetic study in which this analytical method was used. Based on the suggestions of the editor-in-chief, additional information on the analytical methodology was added to the revised version of the manuscript.

Round 2

Reviewer 1 Report

After the 1st revision all my critical comments were addressed. I have no further comments or suggestions for the improvement. I suggest MS aceeptance.

Reviewer 2 Report

Additional information on the analytical methodology which added to the revised version of the manuscript is sufficient and improve the quality of the work. The revised version of the manuscript has better scientific soundness.